# Modular synthesis of PAHs from aryl halides and terminal alkynes via photoinduced palladium catalysis

Chen Zhou[1,2], Pei-Shang Li[1,2] & Ming Chen [1] ✉

A visible-light-induced, palladium-catalyzed one-step annulation of aryl halides with terminal alkynes is developed to enable direct access to polycyclic aromatic hydrocarbons (PAHs) under mild conditions. Unlike conventional thermal methods that favor Sonogashira coupling, this transformation proceeds via photoexcitation of Pd(0), generating a Pd(I)/aryl radical hybrid that adds to the alkyne to form a vinyl radical intermediate, enabling regioselective annulation without requiring oxidative addition. The reaction exhibits broad functional group tolerance, wide substrate scope, and excellent scalability, granting modular entry to diverse PAH frameworks, including π-extended and halogenated derivatives. Mechanistic investigations, including radical trapping, radical clock analysis, EPR experiments, light-dependence experiments, and kinetic isotope effect studies, support a radical pathway. This operationally simple and mechanistically distinct approach provides a streamlined and versatile strategy for the construction of functionalized PAHs with potential relevance to materials science and organic electronics.

Polycyclic aromatic hydrocarbons (PAHs), particularly phenanthrene derivatives, are integral to diverse applications owing to their distinctive electronic, optical, and biological properties[1,2]. Their excellent charge transport and tunable fluorescence make them key components in optoelectronic devices such as OLEDs and OFETs[3,4]. Moreover, PAHs serve as core structural motifs in bioactive molecules, including pharmaceuticals and agrochemicals, underscoring their significance at the interface of medicinal and materials chemistry[5–7]. Despite substantial progress in PAH synthesis[8–13], the development of more efficient, selective, and scalable methods remains a critical and ongoing challenge.

Palladium-catalyzed cross-coupling reactions constitute a cornerstone of modern synthetic chemistry, with far-reaching impact across materials science and pharmaceutical development[14–17]. Since their inception[18], advances in catalyst and ligand design have significantly expanded their mechanistic diversity, enabling transformations beyond the classical oxidative addition–transmetalation–reductive elimination sequence. Among these, the Sonogashira coupling stands out as a powerful method for constructing C(sp)–C(sp²) bonds between terminal alkynes and aryl or alkenyl halides (Fig. 1A)[19–22]. Its high efficiency and broad functional group tolerance make it indispensable for the installation of alkynyl motifs in complex molecules. However, the very robustness of this transformation under conventional thermal conditions also imposes limitations: it overwhelmingly follows the classical cross-coupling manifold, leaving little opportunity for mechanistically distinct or divergent pathways. This intrinsic rigidity has hindered efforts to access structurally complex architectures from aryl halides and terminal alkynes beyond standard coupling products.

Recent advances in photoinduced palladium catalysis[23–29] offer a compelling strategy to overcome the mechanistic rigidity of conventional cross-coupling. Unlike thermally driven processes, photoexcited Pd(0) complexes can undergo single-electron transfer (SET) to generate radical-type intermediates with non-classical reactivity. Notably, such Pd species can serve dually as visible-light absorbers and catalytic centers, eliminating the need for exogenous photocatalysts and unlocking otherwise inaccessible bond-forming pathways[30]. Pioneering work by Gevorgyan[31–35], Fu[36–38], Glorius[39–43], Rueping[44–46], and others[47–51] has demonstrated the power of this strategy in remote

[1]Jiangsu Key Laboratory of Advanced Catalytic Materials & Technology, School of Petrochemical Engineering, Changzhou University, Changzhou, China.
[2]These authors contributed equally: Chen Zhou, Pei-Shang Li. ✉e-mail: chenming0228@cczu.edu.cn

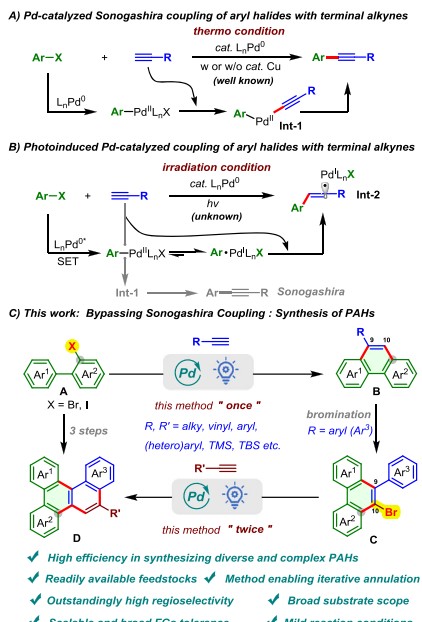

**Fig. 1 | Pd-Catalyzed coupling of aryl halides with terminal alkynes. A** Pd-catalyzed Sonogashira coupling of aryl halides with terminal alkynes. **B** Photoinduced Pd-catalyzed coupling of aryl halides with terminal alkynes. **C** Bypassing Sonogashira Coupling: Synthesis of PAHs.

desaturation, difunctionalization, alkylation, and carbonylation reactions. Among the key species generated in these photoinduced systems are Pd(I)/radical hybrid intermediates. Pd(I)/alkyl radical complexes have been widely exploited, particularly for their reactivity toward olefins[36–38,52–54], dienes[39–42,55–57], and hydrogen atom transfer (HAT) processes[58–60]. In contrast, the chemistry of Pd(I)/aryl hybrid radical species remains relatively underexplored, with known examples primarily involving hydrogen abstraction[23] or intramolecular radical addition to tethered alkenes[43,60–62]. Critically, to the best of our knowledge, no precedents exist for intermolecular engagement of Pd(I)/aryl radical species with terminal alkynes, an elementary transformation that would forge a vinyl radical intermediate (Int-2), opening the door to new annulation strategies (Fig. 1B).

Building on our continued efforts in photoexcited palladium catalysis[62–68], this study addresses a key mechanistic gap—the absence of precedent for productive intermolecular addition of Pd(I)/aryl radical hybrid species to terminal alkynes. We envisioned that photoexcitation of a Pd(0) complex would trigger single-electron transfer (SET) to a 2-halogenated biaryl substrate, generating a Pd(I)/aryl radical intermediate capable of directly engaging terminal alkynes via radical addition. This pathway bypasses the classical oxidative addition–transmetalation–reductive elimination sequence and enables a one-step radical cascade annulation to access phenanthrene scaffolds under mild photochemical conditions (Fig. 1B, C). In addition to its mechanistic features, this photoinduced annulation strategy provides a streamlined and modular route to functionalized phenanthrenes with an open 10-position, facilitating site-selective halogenation and iterative annulations to construct π-extended PAHs. By combining radical reactivity with Pd catalysis, this platform offers a broadly applicable, scalable approach for the synthesis of structurally complex PAHs relevant to advanced organic materials and electronics.

## Results and discussion

### Reaction discovery and mechanistic studies

After extensive optimization, the reaction conditions were established to include Pd(OAc)$_2$ as the catalyst, DPEPhos as the ligand, and K$_3$PO$_4$ as the base, with benzene as the solvent under blue LED irradiation in a

nitrogen atmosphere (see Supplementary Information, S10). Under these conditions, both 2-iodobiphenyl (1a) and 2-bromobiphenyl (1a-Br) underwent efficient coupling with phenylacetylene (2a), affording the annulated product 3a in 82% yield (Fig. 2A). Given the mechanistic ambiguity, specifically, whether product formation arises from a classical Sonogashira coupling followed by 6-endo-dig cyclization, we prioritized elucidation of the reaction pathway. To this end, a series of control and mechanistic experiments were conducted to gain deeper insight into the nature of this transformation.

To determine whether product 3a arises from a classical Sonogashira coupling followed by 6-endo-dig cyclization, the putative intermediate 31 was independently synthesized and subjected to both standard photocatalytic conditions and thermal conditions (80 °C). In both cases, no conversion to 3a was observed, and 31 was recovered in near-quantitative yield, indicating that 6-endo-dig cyclization is disfavored under the reaction conditions (Fig. 2B).

Furthermore, the complete inhibition of product formation in the presence of air suggests a radical-based mechanism. This hypothesis was supported by radical trapping experiments: addition of 1.0 equiv of TEMPO entirely suppressed the reaction, and the corresponding TEMPO–phenyl adduct 37 was detected by HRMS, with no formation of 3a (Fig. 2D). Further evidence was obtained from a radical clock experiment employing a cyclopropyl-substituted styrene. The formation of benzocyclohexene 35 is consistent with a radical-induced ring-opening process followed by either radical cyclization or electrophilic palladation of intermediate 34. In addition, the detection of compound 36, resulting from direct reductive elimination of 34, further confirms the involvement of an aryl radical and supports the radical nature of the transformation (Fig. 2C). This conclusion is further corroborated by EPR spectroscopy: clear signals for the aryl radical–DMPO adduct were observed under the reaction conditions, both in the absence and presence of the terminal alkyne, and were validated by HRMS analysis (Fig. 2F). These direct EPR results provide robust confirmation of the radical pathway operative in this system.

To further probe the reaction mechanism, a light on–off experiment was conducted under standard conditions. Product formation was entirely dependent on continuous light irradiation, with no reaction observed in the dark, indicating that photoactivation is essential and that the transformation does not proceed via a radical chain mechanism (Fig. 2E). Kinetic isotope effect (KIE) studies were also performed to gain further insight. Competitive and parallel experiments using 1a and its deuterated analog 1a–D$^5$ revealed a small isotope effect (K$_H$/K$_D$ = 1.3), indicating that C–H bond cleavage is not involved in the rate-determining step. In contrast, an intramolecular KIE experiment using 2′-deutero-2-iodobiphenyl (1a–D) showed a more pronounced isotope effect (P$_H$/P$_D$ = 1.67), consistent with a normal secondary KIE[68,69]. These results suggest that the rate-limiting step likely involves a change in the hybridization of the carbon bearing the hydrogen or deuterium (sp$^3$ → sp$^2$, as in the conversion from Int-D to 3/4 shown in the mechanistic scheme) rather than direct C–H bond cleavage (Fig. 2G).

Based on these mechanistic studies, a plausible reaction mechanism is proposed (Fig. 2H). The process is initiated by photoexcitation of Pd(0), generating an excited-state species that undergoes single-electron transfer (SET) to aryl iodide 1a to form a Pd(I)/aryl radical hybrid intermediate (Int-A or Int-A′). Under visible-light irradiation, Int-A is the dominant species, while Int-A′ exists in minor amounts and undergoes Sonogashira-type coupling with alkyne 2 under basic conditions to form a byproduct. In contrast, Int-A engages in radical addition to 2, affording vinyl radical intermediates Int-B and Int-B′. From this point, two mechanistic pathways are conceivable. In Path A, the vinyl radical in Int-B undergoes intramolecular *ipso*-cyclization onto the aryl ring, followed by C–H elimination to furnish product 3/4 and regenerate Pd(0). In Path B, Int-B′ undergoes direct electrophilic palladation to form a palladacyclic intermediate (Int-C),

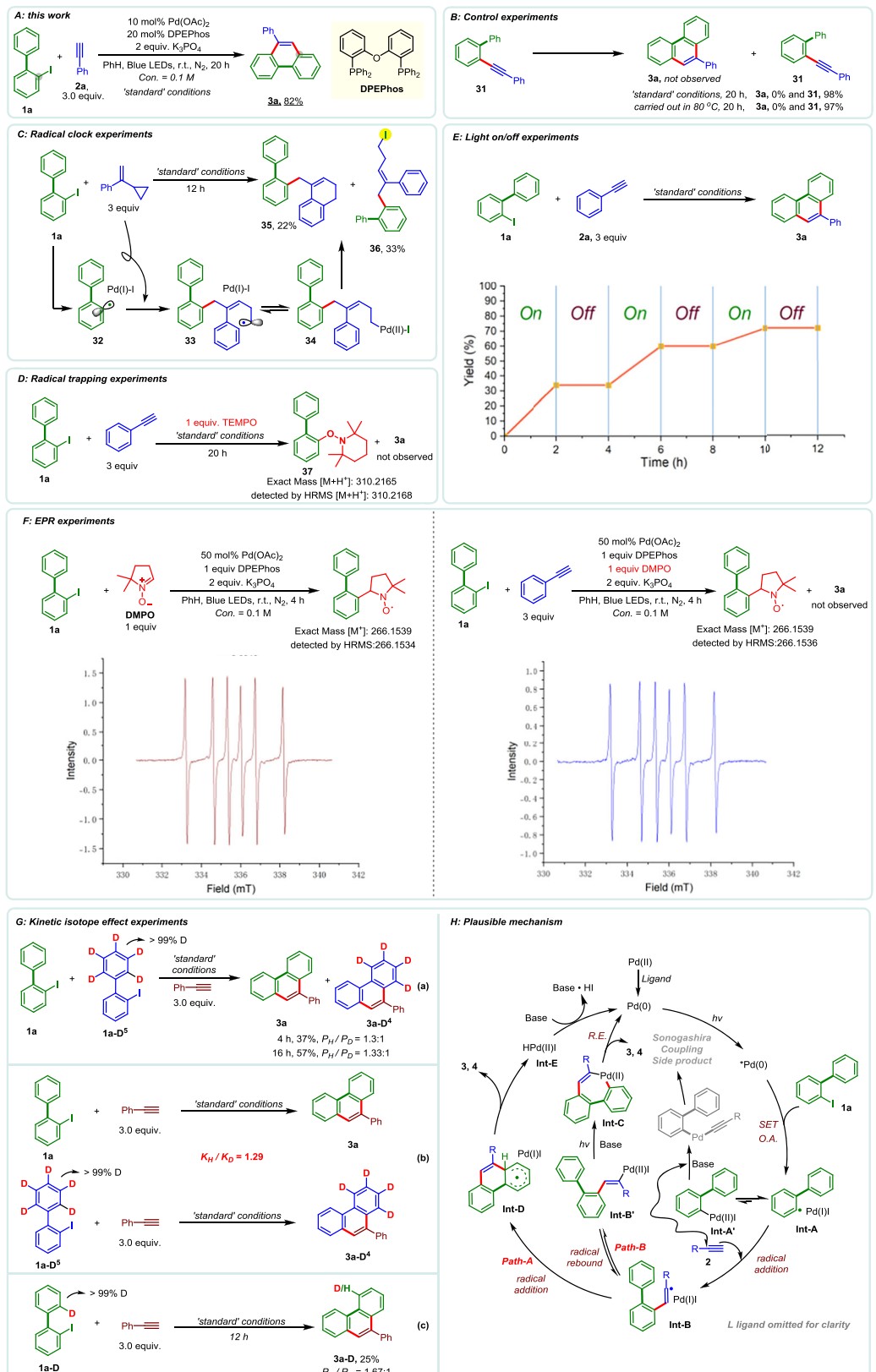

**Fig. 2 | Method development and mechanistic investigations. A** This work. **B** Control experiments. **C** Radical clock experiments. **D** Radical trapping experiments. **E** Light on/off experiments. **F** EPR experiments. **G** Kinetic isotope effect experiments. **H** Plausible mechanism.

which then delivers 3/4 via reductive elimination. Path A is favored for three reasons: (1) electron-donating substituents on the aryl ring accelerate the reaction, consistent with enhanced reactivity of an electron-deficient vinyl radical in cyclization *(vide infra. Scope of 2-iodo-*

*1,1′-biaryl.)*; (2) the intramolecular KIE experiment indicates a hybridization change (sp³ in Int-D→ sp² in 3/4) during product formation, aligning with the C−H elimination event in Path A, and (3) Mechanistically, the radical addition of the Pd(I)/aryl species to the alkyne is

known to proceed via a trans-addition mode, yielding intermediate Int-B'. For Path B to operate, this intermediate would require photo-induced *E/Z* isomerization of the double bond before electrophilic palladation can occur to form Int-C. This conformational rearrangement introduces an additional energy barrier. Nevertheless, given the complexity of the system, the involvement of Path B cannot be entirely ruled out.

## Synthetic application to PAHs construction

With key mechanistic features established, attention turned to demonstrating the broad synthetic utility of this methodology for the efficient construction of polycyclic aromatic hydrocarbons (PAHs). Given the central role of PAHs in organic electronics, materials science, and photonics, access to structurally diverse PAHs remains a pressing synthetic goal. Notably, compounds 3a and 4 s, highlighted in our substrate scope (*vide infra*, *Substrate Scope*), serve as pivotal intermediates in the synthesis of pyrene derivative 29 (Fig. 3D, Eq. (4))[70] and PAH 30 (Fig. 3D, Eq. (5))[71], both of which are integral components in OLED electroluminescent devices[72]. While these examples underscore the relevance of our approach to materials-oriented synthesis, they represent only a fraction of the broader structural space accessible through this strategy.

Despite substantial advances in PAH synthesis, the development of highly efficient and transformative strategies that streamline access to complex PAH frameworks remains a significant challenge. Our methodology addresses this by enabling rapid construction of multi-ring PAHs in a step-economical and operationally simple manner. For example, starting from readily available 1a, a one-step annulation with 2-ethynyl-1,1'-biphenyl affords intermediate 5 in 79% yield. Sequential oxidative cyclization with FeCl₃ delivers dibenzo-[*g, p*]chrysene (6) in 86% yield, followed by a final dehydrogenative cyclization to furnish the fully conjugated PAH 7[73] in three concise steps (Fig. 3A, Eq. (1)). The synthetic utility of this approach is further demonstrated in the iterative expansion of PAH frameworks. Electrophilic halogenation at the electron-rich C10 position of 3q, obtained from a 3 mmol scale reaction of 1a with 2-naphthylacetylene (68%), enables site-selective functionalization. Subsequent bromination with NBS (83%) and a second annulation furnish benzo[*f*]picene (9) in 62% yield (Fig. 3A, Eq. (2)). This three-step sequence provides access to picene derivatives[74–77], notable for their superconductive properties, in an overall 35% yield, highlighting the method's efficiency and scalability. Expanding the modularity of this strategy, scale-up of the annulation between 1a and 1-bromo-2-ethynylbenzene (5 mmol) afforded 9-(2-bromophenyl)phenanthrene (3j) in 45% yield. By Sonogashira coupling and ICl-mediated cyclization furnished extended PAH 11 in 82% over two steps[78,79]. A final annulation step produced chrysene derivative 13, a fully π-extended system featuring seven fused aromatic rings in 73% yield (Fig. 3A, Eq. (3)). This four-step sequence, delivering a structurally elaborate PAH in 27% overall yield, exemplifies the scalability, adaptability, and architectural precision enabled by this methodology.

Collectively, these transformations underscore the exceptional efficiency, modularity, and scalability of our strategy for constructing structurally diverse PAHs. By integrating iterative annulation with strategic functionalization, this approach provides a powerful and flexible platform for the streamlined synthesis of complex PAH frameworks with broad potential in materials science and organic electronics.

Beyond these applications, our methodology enables access to halogenated and unfunctionalized PAHs via a formal "HC≡CH" annulation strategy (Fig. 3B). Notably, even less reactive alkynes such as TMS-acetylene are well tolerated, highlighting the method's adaptability. A key demonstration is the concise synthesis of 9-iodophenanthrene—a valuable synthetic intermediate for C–C[80–84], C–N[85,86], C–O[87,88], C–S[89,90], C–P[91,92], and C–B[93,94] bond formation. Traditional methods require a four-step sequence starting from 1a, involving

Sonogashira coupling, TMS deprotection, iodination, and 6-endo-dig cyclization, with an overall yield of only 33%[95]. In contrast, our approach delivers 9-iodophenanthrene in just two steps and 79% overall yield via annulation of 1a with TMS-acetylene to form tri-methyl(phenanthren-9-yl)silane (15, 87%), followed by ICl-mediated iodination (16, 90%). Moreover, intermediate 15 enables direct access to a range of unfunctionalized PAHs via HOTf-mediated TMS deprotection[96]. This two-step sequence affords phenanthrene (17, 66%), chrysene (18, 68%), tetrahelicene (19, 49%), and a series of benzo-fused PAHs including benzo[*g*]chrysene (20, 69%), benzo[*b*]chrysene (21, 70%), benzo[*c*]chrysene (22, 54%), and benzo[*f*]picene (23, 73%). Compared to conventional PAH syntheses[12,13,95], which often involve harsh conditions or labor-intensive functional group manipulations, this strategy significantly reduces step count while maintaining high efficiency and yield. Collectively, these results highlight the versatility and synthetic economy of this method, establishing it as a valuable tool for scalable access to both functionalized and unfunctionalized PAHs.

In addition to these applications, an intriguing selectivity was observed when 2,2"-diiodo-1,1':4',1"-terphenyl (24) was employed as the substrate (Fig. 3C). In contrast to other substrates, where both aryl iodides could potentially undergo annulation, only one C–I bond cyclized to afford phenanthrene derivatives 25 and 27 in good yields, with the second iodide left intact. This *mono*-selectivity likely stems from steric and conformational constraints imposed by the first annulation, which hinder the spatial reorganization necessary for a second cyclization. Although only one ring closure occurs, this selectivity presents a valuable synthetic opportunity: the unreacted aryl iodide serves as a handle for subsequent cross-coupling reactions. Indeed, Suzuki and Sonogashira couplings proceeded efficiently to deliver highly substituted phenanthrene-based PAHs 26 and 28 in excellent yields. This sequential, site-selective functionalization underscores the modularity and tunability of the methodology.

## Substrate scope with terminal alkynes

After demonstrating the efficiency of this method for PAH construction, we next investigated its substrate scope and functional group compatibility (Fig. 4). A wide range of terminal alkynes reacted efficiently with 2-iodobiphenyl (1a), affording the corresponding phenanthrene derivatives (3) in good to excellent yields, highlighting the versatility and robustness of this protocol. Initial experiments with arylacetylenes revealed broad functional group tolerance. Both electron-donating groups (*n*-butyl (3b), *tert*-butyl (3c), methyl (3d), methoxy (3e), amino (3f)) and electron-withdrawing groups (fluoro (3g), chloro (3h), bromo (3i and 3j), acetyl (3k), trifluoromethyl (3l), cyano (3m), and formyl (3p)) were well tolerated, resulting in good yields of the desired products. Substrates with extended conjugation, such as biphenyl (3n, 3o) and naphthyl (3q) derivatives, also reacted efficiently, further demonstrating the method's broad applicability. Heteroaryl-substituted alkynes, including thiophene (3r), pyridine (3s and 3t), and ferrocene (3u), participated smoothly, affording the corresponding phenanthrene derivatives in moderate to excellent yields. In addition to arylacetylenes, alkenyl- and alkyl-substituted alkynes (3v−3x) also underwent efficient cyclization, with free hydroxyl groups on the alkyl chains tolerated, as seen in the successful conversion of hydroxyl-bearing substrates 3aa and 3ab. The methodology also accommodated bulky or sensitive functional groups, such as TBS-protected alkynes (3y) and ethyl propiolate (3z). Encouragingly, it proved effective with biologically relevant alkynes derived from natural products, including thymol (3ac), cholesterol (3ad), menthol (3ae), vitamin E (3af), and estrone (3ag), delivering the corresponding phenanthrene derivatives in good yields. Overall, these results underscore the broad substrate scope, functional group compatibility, and practical utility of this method for the synthesis of complex molecules. Although some substrates, particularly aryl

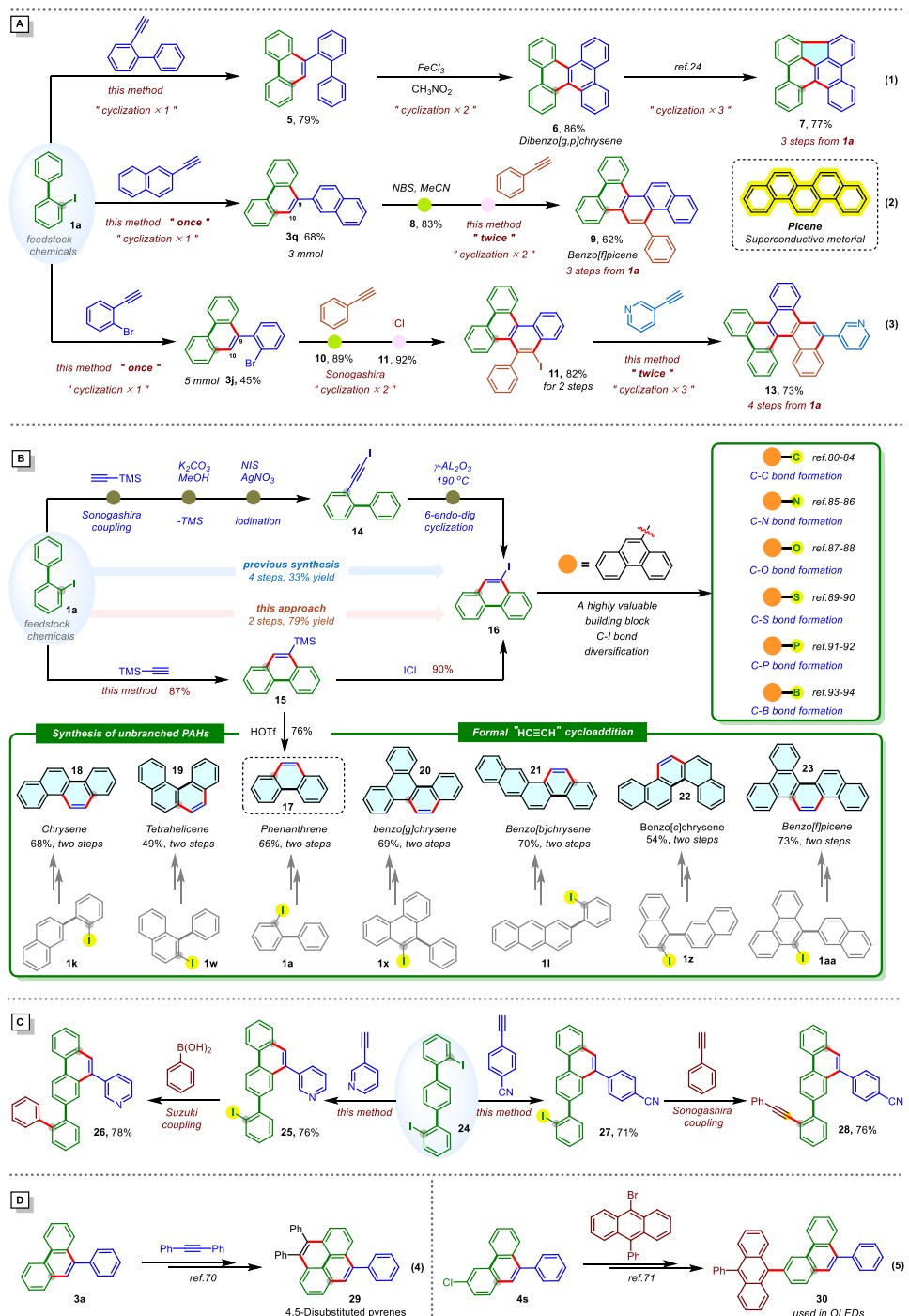

**Fig. 3 | Synthetic utility. A** Synthesis of PAHs. **B** Synthesis of unbranched PAHs. **C** Transformations of 2,2''-Diiodoterphenyl **D** Derivatization of 3a and 4 s to functional materials.

bromides as well as alkyl- and alkenyl-substituted alkynes, showed relatively moderate yields due to minor side reactions (e.g., partial Sonogashira coupling, hydrodehalogenation, or residual starting material), these examples still highlight the broad substrate compatibility of this protocol under unified reaction conditions.

## Substrate scope with 2-iodo-1,1′-biaryls

The generality of this transformation was further explored by evaluating the scope of 2-iodo-1,1′-biaryls (Fig. 5), focusing on variations at both Ar[1] and Ar[2] to assess electronic and steric influences. For Ar[1], a variety of substituents on the aryl ring demonstrated broad functional group

tolerance. Electron-donating groups such as ethyl (**4b**), *tert*-butyl (**4c**), and methoxy (**4d**) afforded moderate to good yields, while electron-withdrawing groups, including chloro (**4f, 4g/4g′**), nitro (**4h/4h′**), and ester (**4j**) were similarly well tolerated. Meta-substituted substrates yielded *regio*-isomeric mixtures (e.g., **4e/4e′**, **4g/4g′, 4h/4h′**), which were readily separable, offering access to distinct phenanthrene isomers despite limited regiocontrol. Remarkably, with extended aromatic systems such as 2-naphthyl and 2-anthryl, exclusive cyclization at the 1-position was observed (**4k, 4l**), likely due to preferential attack of the electron-deficient vinyl radical at the most electron-rich positions within the extended π-system. Heteroaryl-substituted Ar[1] groups including

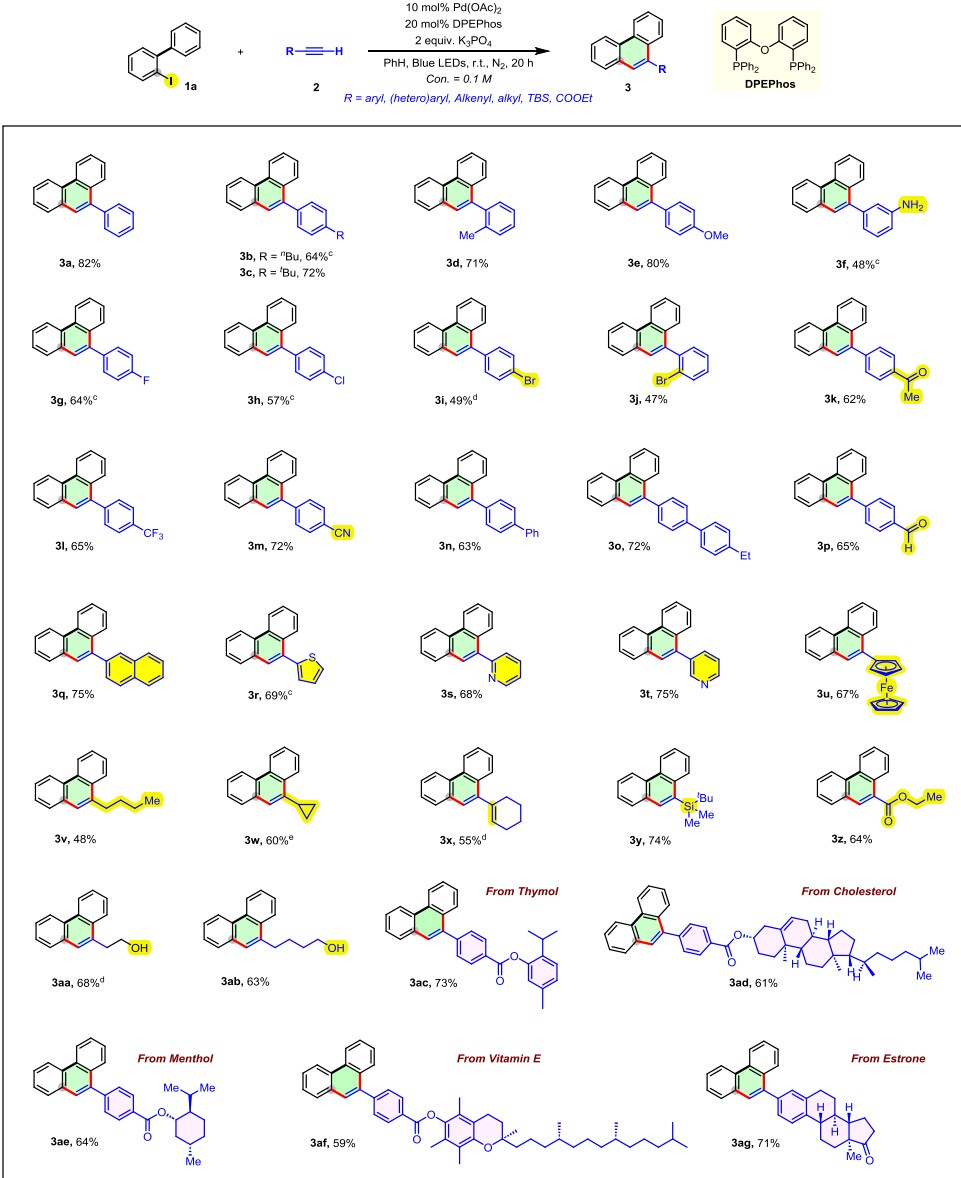

**Fig. 4 | Substrate Scope with terminal alkynes.** [a,b,c,d,e] [a]Reaction conditions: each reaction was run on a 0.2 mmol scale in a sealed 4 mL vial. 2-iodo-1,1'-biphenyl 1a (0.2 mmol, 1.0 equiv), terminal alkyne 2 (0.6 mmol, 3.0 equiv), Pd(OAc)$_2$ (0.02 mmol, 10 mol%), DPEPhos (0.04 mmol, 20 mol%), K$_3$PO$_4$ (0.4 mmol, 2 equiv) and benzene (2 mL), at room temperature. under the irradiation of blue LED lamps for 20 h. [b]Isolated yields. [c]$^t$BuOLi was used as the base. [d]10 mol% (dppf)PdCl$_2$ was used. [e]1a-Br was used.

pyridine (**4m**) and pyrrole (**4n**) were well accommodated, while indole (**4o**) and benzothiophene (**4p**) showed diminished reactivity, reflecting heteroatom-dependent reactivity trends. Evaluation of Ar$^2$ substituents revealed comparable functional group tolerance. Electron-donating and withdrawing groups such as methyl (**4q**), methoxy (**4r**), chloro (**4 s, 4 u, 4 v**), and bromo (**4t**) were efficiently incorporated. Notably, polycyclic aryl groups at Ar$^2$, including naphthalene and phenanthrene, enabled access to structurally extended PAHs such as tetrahelicene-like frameworks (**4w**), benzo[g] chrysene (**4x–4z**), and naphtho[1,2-g]chrysene (**4aa**). When both Ar$^1$ and Ar$^2$ were naphthyl rings, the reaction delivered the highly fused pentacyclic product **4ab** in moderate yield. Together, these results highlight the exceptional scope and functional group compatibility of this annulation strategy. The ability to tolerate a wide range of electronic and sterically demanding biaryl systems underscores the synthetic robustness and modularity of the methodology, offering a scalable platform for accessing structurally diverse phenanthrene-based PAHs.

In summary, a visible-light-driven palladium-catalyzed annulation strategy has been developed for the efficient synthesis of polycyclic aromatic hydrocarbons (PAHs) from 2-halobiaryls and terminal alkynes. This transformation proceeds under mild conditions and circumvents the classical cross-coupling manifold, directly forging fused aromatic frameworks in a single step. The protocol displays broad substrate scope, high functional group tolerance, and excellent scalability, enabling streamlined access to structurally diverse PAHs, including π-extended and halogenated architectures. Iterative annulation sequences further expand the accessible molecular space, allowing modular construction of complex PAH frameworks in a step-economical manner. Mechanistic studies support a radical pathway involving photoexcited Pd(0), formation of Pd(I)/aryl radical hybrids, and selective annulation via vinyl radical intermediates. This strategy offers a general and practical platform for PAH synthesis with broad relevance to materials science and molecular electronics.

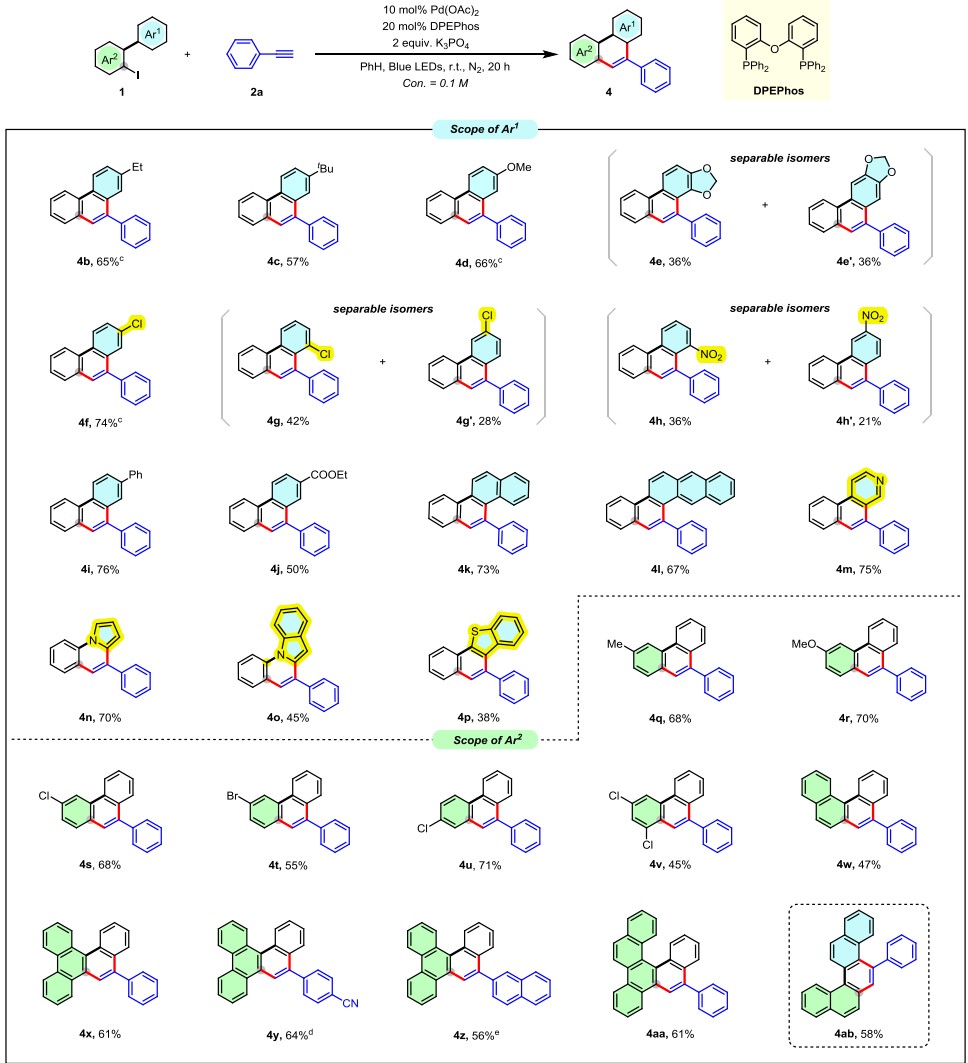

**Fig. 5 | Substrate Scope with 2-iodo-1,1′-biaryls.** [a,b,c,d,e a]Reaction conditions: each reaction was run on a 0.2 mmol scale in a sealed 4 mL vial. 2-iodo-1,1′-biaryl 1 (0.2 mmol, 1.0 equiv), phenylacetylene 2a (0.6 mmol, 3.0 equiv), Pd(OAc)$_2$ (0.02 mmol, 10 mol%), DPEPhos (0.04 mmol, 20 mol%), K$_3$PO$_4$ (0.4 mmol, 2 equiv) and benzene (2 mL), at room temperature. under the irradiation of blue LED lamps for 20 h. [b]Isolated yields. [c]10 mol% (dppf)PdCl$_2$ was used. [d]3 equiv 4-ethynylbenzo nitrile was used. [e]3 equiv 2-ethynylnaphthalene was used.

## Methods

### Typical procedure for the synthesis of product ¾

An oven-dried 4.0 mL vial was charged with 2-aryl iodinated arenes **1** (0.2 mmol, 1.0 equiv.), terminal alkyne **2** (0.6 mmol, 3.0 equiv.), Pd(OAc)$_2$ (4.5 mg, 0.02 mmol, 10 mol%), DPEPhos (21.5 mg, 0.04 mmol, 20 mol%) and K$_3$PO$_4$ (84.8 mg, 0.4 mmol, 2.0 equiv.). It was directly transferred in a nitrogen-filled glovebox with caps. In the glovebox, 2 mL of degassed benzene (PhH) were added to the vial. The vial was tightly sealed, transferred out of the glovebox and stirred at room temperature under the irradiation of blue LEDs lamps for 20 h. After completion of the reaction, the resulting mixture was diluted with acetone (5 mL), filtered (Celite), and concentrated under a reduced pressure. The residue was purified by column chromatography on silica gel (Petroleum ether to Petroleum ether/ Ethyl acetate = 5:1) to afford **3/4**.

## Data availability

All data supporting the findings of this study are available in the Supplementary Information file. These include general experimental procedures, detailed reaction conditions, characterization data, and spectroscopic data (¹H and ¹³C NMR spectra) for all new compounds.

Data supporting the findings of this manuscript are also available from the corresponding author upon request.

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

## Acknowledgments

Changzhou University supports this work for a startup fund (ZMF21020031) and the financial support by NSFC/China (22101034), Changzhou Leading Innovative Talent Project (CQ20210112), Jiangsu specially appointed profes-sors program. We also thank the Analysis and Testing Center, NERC Biomass of Changzhou University for the assistance in NMR analysis.

## Author contributions

M.C. conceived the projects. C.Z. and P.L. performed the experiments under the supervision of M.C. and M.C. wrote the manuscript with the feedbacks of C.Z. and P.L

## Competing interests

The authors declare no competing interests.
