## [Transparent Peer Review file · Nature Communications]

Modular Synthesis of PAHs from Aryl Halides and Terminal Alkynes via Photoinduced Palladium Catalysis

Corresponding Author: Professor Ming Chen

Version 0:

Reviewer comments:

Reviewer #1

(Remarks to the Author)

The development of selective and efficient synthetic methodology for the construction of PAHs is an important content in the field of organic synthetic chemistry because of the versatile applications of PAHs in organic optoelectronic materials and devices. The conventional synthetic routes to these structures typically involve transition metal-catalyzed coupling reactions such as Suzuki coupling and Sonogashira reaction, and following intramolecular cyclization in the presence of oxidants and Lewis acids. In this work, Chen and co-workers developed a visible-light-induced, palladium catalyzed one-step annulation of aryl halides with terminal alkynes, which provides a direct access to diverse PAHs under mild conditions. Mechanistic investigations based on radical trapping experiments indicates the SET nature of this reaction. In addition, a Pd(I)/aryl radical intermediate capable of directly engaging terminal alkynes via radical addition was proposed in the catalytic cycle. This process is different from the conventional palladium-catalyzed Sonogashira reaction under thermal conditions. The reaction exhibits good functional group tolerance and broad substrate scope. Based on these results, I suggest the publication of this manuscript in Nature Communications after addressing the following comments.

- 1) In the investigation of substrate scope, some examples only exhibited moderate yields. In this case, the mass balance is suggested to be elucidated. Side reactions or low reactivities? At least, the results in the representative examples should be given.
- 2) In the mechanistic investigation, the authors claimed that "the intramolecular KIE experiment indicates a hybridization change ($sp^3 \rightarrow sp^2$) during product formation, aligning with the C-H elimination event in Path A". Although ref. 68 and 69 are cited, more detailed explanation is suggested. As a reader, I could not well understand this inference.
- 3) The examples in this work are typically based on mono-iodo biaryls. How about diiodides. Could double annulation be realized under the photoexcited palladium catalysis. Highly π -extended PAHs could be more attractive.
- 4) Some typo mistakes in the SI: for example, Page S40, " ^{13}C NMR (101 MHz, $CDCl_3$) ^{13}C NMR (101 MHz, $CDCl_3$)".

Reviewer #2

(Remarks to the Author)

This manuscript by Ming Chen and co-workers developed a mild method to Synthesis of PAHs from Aryl Halides and Terminal Alkynes via Photoinduced Palladium Catalysis. Unlike conventional thermal methods that favor Sonogashira coupling, this transformation proceeds via photoexcitation of Pd(0), generating a Pd(I)/aryl radical hybrid that adds to the alkyne to form a vinyl radical intermediate, enabling regioselective annulation without requiring oxidative addition to access polycyclic aromatic hydrocarbons (PAHs) under mild conditions. This method exhibits broad functional group tolerance, wide substrate scope, and excellent scalability. As such, this reviewer recommends the publication of the work in nature communication after minor revisions indicated below:

1. The authors did a series of mechanistic control experiments to indirectly demonstrate that the reaction proceeds via a radical pathway. EPR experiments are highly desirable to directly confirm the radical pathway.
2. After a multi-step reaction to achieve the conversion from 1a to 9/13, how did the authors determine the structure of 9/13, and if it is a known compound, please cite the relevant literature. Did the authors monitor the cyclization product to another aromatic ring?
3. In addition to the above substrates, please test whether substrates similar to the structure of 2'-iodo-1,2,3,4-tetrahydro-1,1'-biphenyl are suitable for this system.

4. Some substrates require further purification, such as 13.

Reviewer #3

(Remarks to the Author)

The manuscript by Chen and coworkers reports a palladium-catalyzed single-step annulation between aryl halides and terminal alkynes, achieving direct synthesis of polycyclic aromatic hydrocarbons via visible-light-induced intermolecular cyclization. This work carried out in-depth mechanistic studies and synthesized a variety of polycyclic aromatic hydrocarbons. Although intramolecular cyclization synthesis of PAHs has been widely reported (J. Org. Chem. 2025, 90, 2230; J. Am. Chem. Soc. 2021, 143, 15420; Patent CN117164429A, 2023; Org. Lett. 2004, 6, 2677), this intermolecular light-induced cyclization of non-Sonogashira Coupling is innovative. Consequently, I recommend the publication of this article in Nature Communications after the revisions listed below.

(1) Methods for synthesizing analogous PAHs have been widely reported (J. Org. Chem. 2025, 90, 2230; J. Am. Chem. Soc. 2021, 143, 15420; Patent CN117164429A, 2023; Org. Lett. 2004, 6, 2677). It is recommended that more tests be performed to evaluate their potential applications, such as synthesizing the molecules having in liquid crystal material properties.

(2) Relevant papers on intramolecular cyclization "J. Org. Chem. 2025, 90, 2230; Org. Lett. 2004, 6, 2677" should be cited.

(3) Please calibrate the yield of the 3n system. If the experimental results indicated that the yield remained low, it would be desirable to analyze the remaining starting materials and by-products in the system.

(4) In the expansion of heterocyclic substrates, attempts involving oxygen-containing aromatic heterocycles (e.g., oxa-aromatics) were notably absent.

(5) Did the authors attempt to employ acetylene as a reactant in the synthetic preparation?

(6) The ¹H NMR spectra of compounds 3a, 3c, 3e, 3f, 3aa, 8, 9, 27, and 35 are not clean. The products should be purified and the spectra should be measured again in order to remove solvent peaks or impurity signals. The ¹³C NMR spectra of compounds 3f, 3k, 3v, and 27, the signal-to-noise ratio (S/N) needs to be improved.

(7) The authors mainly used aryl iodides throughout the entire study, with only compound 1w being an aryl bromide. How did the reaction perform when using aryl iodides in this case? Were there any cyclopropane ring-opening reaction products observed?

(8) Benzene as a solvent poses significant hazards. Please emphasize these risks in the Supporting Information (SI) and summarize standardized protocols for the post-reaction handling procedures.

(9) For mechanistic understanding, it would be useful to prepare a normal Sonogashira coupling product and then subject it to the current photo-induced reaction conditions. This can rule out the possibility that the reaction follows a pathway involving direct Sonogashira coupling followed by further cyclization.

Reviewer #4

(Remarks to the Author)

Version 1:

Reviewer comments:

Reviewer #1

(Remarks to the Author)

The revised manuscript is well organized. All my concerns have been addressed in the revised manuscript and SI. I recommend the publication of this manuscript on Nature Communications.

Reviewer #2

(Remarks to the Author)

The authors have properly addressed the raised issues, which solidified the publication of this manuscript in this journal.

Reviewer #3

(Remarks to the Author)

In the revised manuscript, the authors have addressed all my questions. I feel this work can be published in Nature Communications as it is.

Reviewer #4

(Remarks to the Author)

I co-reviewed this manuscript with one of the reviewers who provided the listed reports. This is part of the Nature Communications initiative to facilitate training in peer review and to provide appropriate recognition for Early Career

Researchers who co-review manuscripts.

CHANGZHOU UNIVERSITY
School of Petrochemical Engineering
21 Gehu Road, Changzhou 213164
People's Republic of China

Ming Chen, Ph.D.
Professor of Chemistry
Changzhou University
Email: chenming0228@cczu.edu.cn
Phone: +86-13961183374

June 20, 2025

We would like to thank the reviewers for their consideration of our manuscript (NCOMMS-25-29016) for publication in Nature Communications.

Here are our replies to these constructive comments:

I. Re: referee 1

1) Original comments: *In the investigation of substrate scope, some examples only exhibited moderate yields. In this case, the mass balance is suggested to be elucidated. Side reactions or low reactivities? At least, the results in the representative examples should be given.*

We sincerely appreciate the reviewer 1's thoughtful comment regarding the moderate yields observed for some examples in Figure 4. As noted, the relatively lower yields are primarily due to minor side reactions, which include small amounts of Sonogashira-type coupling, residual starting materials (particularly for aryl bromide substrates as well as alkyl- and alkenyl-substituted alkynes), hydrodehalogenation products, and trace arylation of the solvent (benzene). These byproducts were identified during our condition optimization (see Figure 2, SP1-SP3) and were also briefly discussed in the manuscript. For entries with lower yields, we did not pursue further condition refinement because our focus was to demonstrate the broad substrate

tolerance under unified standard conditions. We have now added clarifying remarks in the revised manuscript to better explain this point and provide representative examples of the observed side products. We hope this addresses the reviewer's concern satisfactorily.

2) **Original comments:** *In the mechanistic investigation, the authors claimed that "the intramolecular KIE experiment indicates a hybridization change ($sp^3 \rightarrow sp^2$) during product formation, aligning with the C-H elimination event in Path A". Although ref. 68 and 69 are cited, more detailed explanation is suggested. As a reader, I could not well understand this inference.*

We thank the reviewer 1 for pointing out the need for a clearer explanation regarding the intramolecular kinetic isotope effect (KIE) and its mechanistic implications. As described in the manuscript and illustrated in Figure 2F and the mechanistic scheme above, our measured intramolecular KIE value (PH/PD = 1.67) falls within the range of a normal secondary isotope effect. Such a secondary KIE arises because the reaction step involves a change in the hybridization of the carbon atom bonded to the hydrogen (or deuterium) without direct bond cleavage. Specifically, in our system, this corresponds to the transformation from the sp^3 -hybridized intermediate (Int-D) to the sp^2 -hybridized carbon in the annulated product (3/4). This hybridization change is consistent with a C-H elimination step and thus supports Path A as the dominant pathway.

Additionally, we favor Path A for a further stereoelectronic reason: the radical addition of the Pd(I)/aryl species to the alkyne is known to proceed via a trans-addition mode, yielding intermediate Int-B'. To follow Path B, this intermediate would need to undergo a light-driven double bond isomerization to form Int-B'' before electrophilic palladation could occur to reach Int-C. This required E/Z isomerization adds an additional energetic barrier, making Path B less favorable under our reaction conditions. In contrast, Path A does not require such an isomerization and is therefore

more plausible (Scheme 1). We have clarified this discussion in the revised manuscript to help readers better understand this inference.

We appreciate the reviewer 1's valuable suggestion, which has helped us improve the clarity of our mechanistic rationale.

Scheme 1

3) **Original comments:** *The examples in this work are typically based on mono-iodo biaryls. How about diiodides. Could double annulation be realized under the photoexcited palladium catalysis. Highly π -extended PAHs could be more attractive.*

We sincerely appreciate the reviewer's constructive suggestion to explore diiodobiaryl substrates for potential double annulation, which could indeed provide a concise route to highly π -extended PAHs. We share the same interest and accordingly investigated representative diiodo substrates under our optimized photoexcited palladium-catalyzed conditions (see the figure above). Unfortunately, in both cases (using 3 and 6 equivalents of the terminal alkyne), the reactions resulted in complex mixtures without clean formation of the desired double-annulated products, as evidenced by TLC analysis. These outcomes suggest that our current catalytic system is not suitable for effective sequential annulation with diiodoarenes, likely due to increased steric hindrance and competing side reactions after the first ring closure. Nevertheless, we

agree that developing a robust double annulation protocol remains an attractive goal for future study.

Scheme 2

4) Original comments: *Some typo mistakes in the SI: for example, Page S40, “¹³C NMR (101 MHz, CDCl₃) ¹³C NMR (101 MHz, CDCl₃)”.*

We thank the reviewer 1 for carefully pointing out the typo in the Supporting Information (e.g., the repeated “¹³C NMR” on Page S40). We have thoroughly checked and corrected this and other minor inconsistencies in the revised SI. We appreciate the reviewer 1’s attention to detail.

II. Re: referee 2 publish in *Nature Communication* after minor revisions.

1) Original comments: *The authors did a series of mechanistic control experiments to indirectly demonstrate that the reaction proceeds via a radical pathway. EPR experiments are highly desirable to directly confirm the radical pathway.*

We thank the reviewer 2 for this valuable suggestion to directly confirm the radical pathway by EPR spectroscopy. In response, we have performed EPR experiments under our reaction conditions both in the absence and in the presence of the terminal alkyne (Scheme 3). In both cases, a clear EPR signal was observed, indicating the formation of an aryl radical species, which was efficiently trapped by DMPO and further confirmed by HRMS analysis. Importantly, when the alkyne was present, the same aryl radical-DMPO adduct was detected, while no annulation product (3a) was formed under the trapping conditions. These results provide direct and robust evidence supporting the radical nature of our transformation. We have added this EPR data to the revised SI and discussed it in the manuscript. We sincerely appreciate the reviewer 2's helpful recommendation.

Scheme 3

2) Original comments: After a multi-step reaction to achieve the conversion from 1a to 9/13, how did the authors determine the structure of 9/13, and if it is a known compound, please cite the relevant literature. Did the authors monitor the cyclization product to another aromatic ring?

We sincerely thank the reviewer² for this insightful comment regarding the structural confirmation of compounds 9 and 13, and the possibility of alternative cyclization pathways.

For compound 9, it was obtained by annulation of intermediate 8 with phenylacetylene under our standard conditions. The 2-naphthyl moiety indeed provides two potential cyclization sites (1-position and 2-position) (Scheme 4). However, the reaction predominantly proceeds via cyclization at the 1-position, consistent with the regioselectivity observed for similar substrates in our scope (see 4k, eq. 1), which is a known compound and serves as a reliable reference. If cyclization occurred at the 2-position instead, it would lead to the alternative regioisomer 9', which would exhibit two distinct downfield singlets in its ¹H NMR spectrum due to the characteristic protons in that structure. However, such signals were not observed experimentally, supporting our structural assignment for 9.

For compound 13, its precursor 12 indeed has two possible reactive sites. We initially verified the structure by analyzing the ¹³C NMR spectrum: if the undesired regioisomer 13' were formed, the free phenyl ring would lead to two double-height peaks due to equivalent carbons—this was not observed. Additionally, we attempted to obtain single-crystal X-ray data for unambiguous confirmation, but the polycyclic aromatic nature of this compound made crystal growth challenging due to its limited stability. To address this point further and to validate the overall structural assignment, we designed an alternative synthetic route (eq. 4). In this pathway, compound 15 (derived from 3j via Sonogashira coupling and ICl-mediated cyclization) inherently possesses only one reactive site, which undergoes the final annulation to afford 17 unambiguously. Notably, intermediates 3j, 14, and 16 are all known compounds, and the structure of 15 was confirmed via its successful Suzuki coupling to give 16. Thus, this sequence provides additional support for the structural integrity and regioselectivity of our PAH targets, including 9 and 17.

We have added clarifying explanations in the revised manuscript and hope this addresses the reviewer's concern satisfactorily.

Scheme 4

3) Original comments: *In addition to the above substrates, please test whether substrates similar to the structure of 2'-iodo-1,2,3,4-tetrahydro-1,1'-biphenyl are suitable for this system.*

We sincerely thank the reviewer for this valuable suggestion to test substrates analogous to 2'-iodo-1,2,3,4-tetrahydro-1,1'-biphenyl in our system. As shown in Scheme 5, we prepared the corresponding substrate following a straightforward two-

step procedure. However, under our standard photoexcited palladium-catalyzed conditions, the reaction resulted in a complex mixture without formation of the desired annulated product.

We believe this outcome is likely due to undesired intramolecular hydrogen atom transfer (HAT) processes: the aryl radical generated under the reaction conditions may abstract a hydrogen atom from the cyclohexyl moiety, leading to side reactions and decomposition. This hypothesis is supported by GC-MS analysis of the crude mixture, which showed no evidence of the target product.

We appreciate the reviewer's constructive suggestion, which has helped us further understand the limitations of our system. We have included this observation and explanation in the Supporting Information under the section of unsuccessful examples.

Scheme 5

4) Original comments: *Some substrates require further purification, such as 13.*

We thank the reviewer 2 for carefully checking the spectral data. Although compound 13 had already been replaced with a newly prepared batch to ensure structural accuracy, we fully agree with the reviewer's suggestion and took this opportunity to further confirm its purity. We note that a similar concern was also raised by Reviewer 3 regarding the purity and spectral clarity of certain compounds. In response, we have re-purified the relevant compounds and updated their ^1H and ^{13}C NMR spectra to improve data quality. The revised spectra have been included in the updated Supporting Information. We sincerely appreciate the reviewers' comments, which helped us improve the overall rigor of the manuscript.

III Re: referee 3

1) Original comments: *Methods for synthesizing analogous PAHs have been widely reported (J. Org. Chem. 2025, 90, 2230; J. Am. Chem. Soc. 2021, 143, 15420; Patent CN117164429A, 2023; Org. Lett. 2004, 6, 2677). It is recommended that more tests be performed to evaluate their potential applications, such as synthesizing the molecules having in liquid crystal material properties.*

We sincerely thank the reviewer 3 for this thoughtful suggestion regarding potential applications in liquid crystal materials. We have carefully discussed this idea with colleagues at our university who specialize in liquid crystal research and also reviewed relevant literature. As a general design principle, molecules used for liquid crystal applications typically require a significant aspect ratio (length-to-width) and a certain degree of linearity or rod-like shape to promote mesophase formation. However, the PAHs synthesized by our method predominantly possess rigid, fused polycyclic frameworks with relatively low aspect ratios and compact shapes, making them less suitable as liquid crystal mesogens.

Nevertheless, we fully agree that expanding the utility of our method to access functionalized PAHs with tailored geometries for specific material properties is an exciting direction. We hope that in the future, researchers working on advanced functional materials may apply our synthetic strategy to design PAHs better suited for liquid crystal or other optoelectronic applications.

We appreciate the reviewer 3's insightful perspective, which has inspired us to consider broader applications of this methodology in future collaborations.

2) Original comments: *Relevant papers on intramolecular cyclization "J. Org. Chem. 2025, 90, 2230; Org. Lett. 2004, 6, 2677" should be cited.*

We thank the reviewer 3 for this helpful suggestion. We have now cited the recommended papers on intramolecular cyclization as *Ref. 78-79* in the revised manuscript.

3) Original comments: *Please calibrate the yield of the 3n system. If the experimental results indicated that the yield remained low, it would be desirable to analyze the remaining starting materials and by-products in the system.*

We sincerely thank the reviewer 3 for highlighting the yield calibration and mass balance for the 3n system. As noted in our earlier response, the somewhat moderate yield for this substrate, as well as for certain other entries, is mainly attributed to minor side reactions and incomplete conversion under the unified standard conditions. Specifically, we observed small amounts of Sonogashira-type byproducts, residual starting material—particularly when using aryl bromide or alkyl/alkenyl alkynes—and trace hydrodehalogenation and solvent-derived arylation products. These species were detected during our reaction optimization (see Figure 2, SP1-SP3) and have been briefly discussed in the revised manuscript.

For practical reasons, we chose not to further optimize each low-yielding entry individually, focusing instead on demonstrating the method's broad substrate compatibility under consistent conditions. To address this point, we have added clarifying comments and representative examples of the observed side products in the the main text. We hope this explanation clarifies the mass balance and the source of the moderate yield for 3n and related substrates.

4) **Original comments:** *In the expansion of heterocyclic substrates, attempts involving oxygen-containing aromatic heterocycles (e.g., oxa-aromatics) were notably absent.*

We sincerely thank the reviewer for this constructive suggestion regarding the exploration of oxygen-containing aromatic heterocycles (e.g., oxa-aromatics) as substrates in our system. To address this, we synthesized both benzofuran and furan-derived substrates, as shown in Scheme 6 above. However, under our standard conditions, the key annulation reactions did not proceed efficiently: for the benzofuran derivative, only moderate yields of undesired byproducts were observed alongside unreacted starting material; for the furan derivative, the reaction produced a complex mixture without the desired product, as indicated by TLC analysis. We speculate that this outcome is likely due to the mismatch in orbital orientation and steric approach between the radical intermediate formed after alkyne addition and the heteroaromatic ring system of benzofuran or furan. This misalignment may slow down or prevent the critical cyclization step, resulting in side reactions or decomposition. We greatly appreciate the reviewer's valuable suggestion, which has helped us better understand the limitations and scope boundaries of our methodology. We have included this observation and explanation in the Supporting Information under the section of unsuccessful examples.

start material synthesis

The key reaction:

Scheme 6

5) Original comments: *Did the authors attempt to employ acetylene as a reactant in the synthetic preparation?*

We sincerely thank the reviewer 3 for this insightful question regarding the use of acetylene gas as a reactant in our synthetic system. To address this point, we performed

a dedicated experiment using a balloon of acetylene under our standard reaction conditions, as shown in Scheme 7. In this experiment, we observed that most of the starting material (2-iodobiphenyl) was recovered (88% isolated), and only a trace amount (<5%) of the desired phenanthrene product was detected by ¹H NMR analysis of the crude mixture. We attribute this low conversion mainly to the poor solubility of acetylene gas in the organic solvent under these conditions.

Nevertheless, the detection of trace product clearly demonstrates that the method is mechanistically feasible for acetylene as well. We appreciate the reviewer 3's suggestion, which highlights an interesting aspect for further optimization.

Scheme 7

6) Original comments: *The ¹H NMR spectra of compounds 3a, 3c, 3e, 3f, 3aa, 8, 9, 27, and 35 are not clean. The products should be purified and the spectra should be measured again in order to remove solvent peaks or impurity signals. The ¹³C NMR spectra of compounds 3f, 3k, 3v, and 27, the signal-to-noise ratio (S/N) needs to be improved.*

We sincerely thank the reviewer for carefully pointing out the issues with the ¹H and ¹³C NMR spectra of certain compounds. In response, we have re-purified compounds 3a, 3c, 3e, 3f, 3aa, 8, 9, 27, and 35, and re-measured their ¹H NMR spectra to eliminate residual solvent and impurity peaks. Additionally, the ¹³C NMR spectra of 3f, 3k, 3v, and 27 have been re-recorded to improve their signal-to-noise ratio. The revised spectra are provided in the updated Supporting Information. We appreciate the reviewer's constructive feedback, which has improved the data quality.

7) Original comments: *The authors mainly used aryl iodides throughout the entire study, with only compound 1w being an aryl bromide. How did the reaction perform when using aryl iodides in this case? Were there any cyclopropane ring-opening reaction products observed?*

We sincerely thank the reviewer for this careful observation and thoughtful question regarding the use of aryl bromides versus aryl iodides, especially for the cyclopropylacetylene substrate (1w). As noted, for alkyl- and alkenyl-substituted alkynes in general, the reaction efficiency under our standard conditions is relatively lower than for arylacetylenes. Therefore, for these challenging cases, we tested both aryl iodides and aryl bromides to identify the better performing substrate.

Specifically, for the cyclopropylacetylene example, we found that the aryl bromide afforded a noticeably higher yield of the desired annulated product compared to the corresponding aryl iodide (which gave a moderate yield of 45%). To present the best result in the scope, we thus chose to report the aryl bromide variant in this case.

Regarding the cyclopropane unit, we did not observe any evidence of cyclopropane ring-opening products in the crude reaction mixture. This is likely because the key

intermediate in this case is an alkenyl radical; if the cyclopropane were to undergo ring opening, it would generate an allenyl-type radical, which may not be sufficiently stabilized to undergo the subsequent annulation step efficiently. As a result, no ring-opened byproducts were detected under our reaction conditions.

8) Original comments: *Benzene as a solvent poses significant hazards. Please emphasize these risks in the Supporting Information (SI) and summarize standardized protocols for the post-reaction handling procedures.*

We sincerely thank the reviewer 3 for this important comment regarding the use of benzene as a solvent and the associated safety considerations. We fully acknowledge that benzene is classified as a carcinogenic and toxic solvent, and we are therefore committed to handling it with strict precautions.

In our study, only small-scale reactions (typically 2 mL of benzene per 0.2 mmol reaction) were used. All manipulations involving benzene were performed inside a well-ventilated fume hood or a nitrogen-filled glovebox to minimize exposure. After the reaction, the crude mixture was concentrated under reduced pressure using a rotary evaporator, with the exhaust line directed into an active fume hood to ensure that any residual benzene vapors were safely vented.

In response to the reviewer's suggestion, we have included a clear note in the Supporting Information (Page S4) explicitly emphasizing the hazards of benzene, along with recommended best practices for safe handling, solvent removal, and waste disposal according to standard laboratory safety protocols and institutional guidelines. We appreciate the reviewer 3's careful attention to laboratory safety and believe these additions further strengthen the safety documentation for our work.

9) Original comments: *For mechanistic understanding, it would be useful to prepare a normal Sonogashira coupling product and then subject it to the current photo-induced reaction conditions.*

This can rule out the possibility that the reaction follows a pathway involving direct Sonogashira coupling followed by further cyclization.

We sincerely thank the reviewer 3 for this valuable mechanistic suggestion. As detailed in the main text (Figure 2B) and shown below, we had already conducted this control experiment: the isolated Sonogashira coupling product (31) was subjected to our standard photoinduced reaction conditions and also to thermal conditions at 80 °C. In both cases, no annulated product (3a) was formed, and the starting material was recovered almost quantitatively.

These results confirm that the reaction does not proceed via a stepwise Sonogashira coupling followed by cyclization but instead follows the direct radical annulation pathway as proposed. We apologize if this point was not sufficiently clear in the initial manuscript and appreciate the reviewer's attention to this detail.

Scheme 8

Finally, We greatly thank your consideration of this Communication for publication in *Nature Communications* and hope that it meets your high standards for publication.

Sincerely,

Ming Chen

Ming Chen